# Antigenic Site Variation in the Hemagglutinin of Pandemic Influenza A(H1N1)pdm09 Viruses between 2009–2017 in Ukraine

**DOI:** 10.3390/pathogens8040194

**Published:** 2019-10-18

**Authors:** Oksana Zolotarova, Iryna Budzanivska, Liudmyla Leibenko, Larysa Radchenko, Alla Mironenko

**Affiliations:** 1Educational Scientific Centre “Institute of Biology and Medicine”, Taras Shevchenko National University of Kyiv, 01601 Kyiv, Ukraine; birishechka68@gmail.com; 2“Gromashevsky L.V. Institute of epidemiology and infectious diseases, National Academy of Medical Sciences of Ukraine” 03680 Kyiv, Ukraine; liudmyla.leib@gmail.com (L.L.); larysa_rad@ukr.net (L.R.); miralla@ukr.net (A.M.)

**Keywords:** genetic analysis, influenza A(H1N1)pdm09 virus, antigenic site, mutation

## Abstract

The hemagglutinin (HA) is a major influenza virus antigen, which, once recognized by antibodies and substitutions in HA genes, helps virus in escaping the human immune response. It is therefore critical to perform genetic and phylogenetic analysis of HA in circulating influenza viruses. We performed phylogenetic and genetic analysis of isolates from Ukraine, the vaccine strain and reference strains were used to phylogenetically identify trends in mutation locations and substitutions. Ukrainian isolates were collected between 2009–2017 and clustered in the influenza genetic groups 2, 6, 7, and 8. Genetic changes were observed in each of the antigenic sites: Sa – S162T, K163Q, K163I; Sb – S185T, A186T, S190G, S190R; Ca1 – S203T, R205K, E235V, E235D, S236P; Ca2 – P137H, H138R, A141T, D222G, D222N; Cb – A73S, S74R, S74N. In spite of detected mutations in antigenic sites, Ukrainian isolates retained similarity to the vaccine strain A/California/07/09 circulated during 2009–2017. However, WHO recommended a new vaccine strain A/Michigan/45/2015 for the Southern Hemisphere after the emergence of the new genetic groups 6B.1 and 6B.2. Our study demonstrated genetic variability of HA protein of A(H1N1)pdm09 viruses isolated in 2009–2017 in Ukraine. Influenza surveillance is very important for understanding epidemiological situations.

## 1. Introduction

The pandemic influenza virus A(H1N1)pdm09 emerged in the human population in the spring of 2009 and caused a serious pandemic [1]. The emergence of a new highly virulent virus in the human population can cause pandemics, thus, it is important to conduct epidemiological, genetic and antigenic analyses [2]. The variability of influenza viruses is caused by changes in the sequences of their eight genes (PB2, PB1, PA, HA, NP, NA, M, NS). Mutations are the reason for genes oligonucleotide sequence changes of epidemic influenza viruses in the interpandemic period. In turn, the accumulation of amino acid substitutions leads to changes in the antigenic properties of a virus [2,3]. The antigenic properties of hemagglutinin and neuraminidase change rapidly in order to escape neutralization by human antibodies. Mainly because of an antigenic drift, WHO revises the vaccine composition on a yearly basis to include only actual circulating strains and to develop effective vaccines [3]. In addition, a number of cases of human infection with zoonotic influenza viruses (A(H5N1), A(H7N9) ect.)—which may cause severe disease and have potential risk to cause pandemic [4]—were reported.

Antibodies, that recognize hemagglutinin (HA) and neuraminidase (NA) glycoproteins, serve as a main human immune response against influenza virus infection [3,5]. The HA is a trimer glycoprotein on the virus surface, which is composed by three monomers. Each monomer is the HA inactivated precursor, HA0, which is cleaved by host proteases in the human respiratory tract into HA1 and HA2 subunits [4]. Hemagglutinin is a major influenza virus antigen recognized by neutralizing antibodies that inhibit its binding with the cell receptors [6]. HA is susceptible to a rapid evolution in the specific antigenic sites, however, it also has highly conserved regions responsible for the functions important for replication—the binding with a receptor (sialic acid) and the fusion of membranes. Therefore, HA can accumulate amino acid substitutions, leading to an evolution of influenza virus antigenic properties without changes in the main structural sites [7].

The H1 molecule of hemagglutinin has five antigenic sites: Sa, Sb, Ca1, Ca2, and Cb, which are the most variable in the viruses and are recognized by specific antibodies since the emergence of H1N1 viruses [8,9]. Sa and Sb have the most diverse amino acids in the strain-specific antigenic sites, some of them are located near the receptor-binding site (RBS) [10]. The substitutions of amino acids in HA antigenic sites may occasionally lead to the acquisition of side carbohydrate chains [3,11]. When the carbohydrate chains are located close to the antigenic sites, they mask neutralizing epitopes on the HA molecule. The amino acid substitutions associated with the acquiring of the carbohydrate chain effectively generate the emergence of new antigenic variants of viruses [12,13].

Phylogenetic analysis revealed reasortment of A(H1N1)pdm09 in Ukrainian isolates since starting in 2009. Between 2009 and 2017, several genetic changes resulted in the alteration of antigenic properties of influenza viruses. The viruses circulating worldwide at that time belonged to 8 genetic groups. According to the phylogenetic analysis performed in this study, Ukrainian isolates within the mentioned period belonged to the genetic groups 2, 6, 7, and 8 [14]. Beginning in the 2010–2011 season, group 6 viruses predominantly circulated both in Ukraine and worldwide [14]. Group 6 viruses are divided into subgroups 6A, 6B, and 6C, which are characterized by specific mutations. Influenza viruses from genetic group 6B have become the most widespread [14].

In the 2015–2016 season, viruses of genetic group 6B were split into subgroups 6B.1 and 6B.2 due to new substitutions in recently emerged viruses. These substitutions influenced antigenic characteristics as well. Genetic group 6B.1 viruses are the most widespread in the recent epidemic seasons and were detected in the majority of worldwide influenza cases [15]. Characteristic substitutions for the group 6B.1 include point mutations S84N, S162N (+CHO) and I216T in HA1. The specific feature of this group is the acquisition of an additional glycosylation site in position 162, which can potentially increase virulence [16]. The S162N mutation confers a potential gain of glycosylation at residues 162–164 that may mask K163Q and other epitopes relevant for neutralizing the antibody binding [17]. Moreover, position 162 is located in the antigenic site, which can influence the antigenic properties of viruses of this group [11].

Phylogenetic and genetic analyses were performed on Ukrainian pandemic influenza virus isolates collected between 2009–2017 to identify trends in mutation locations and amino acid substitutions.

## 2. Results

### 2.1. Genetic Group Separation

We analyzed 136 isolates, including the vaccine strain A/California/07/2009 and several reference strains from each genetic group. According to the results of phylogenetic analysis, viruses were grouped into separate clusters derived from unique substitutions, as shown in Table 1. Overall, we identified isolates belonging to the 8 established genetic groups of the A(H1N1)pdm09 pandemic influenza. These groups differ from the NIMR [14] vaccine strain A/California/07/2009 due to the specific amino acid substitutions in HA1 and HA2 subunits.

Viruses circulating in Ukraine throughout the research period belonged to genetic groups 2, 6 (6A, 6B, 6B.1, 6B.2, 6C), 7 and 8 as shown in Figure 1. Acquired amino acid substitutions, N31D and A186T, A/Dnipropetrovsk/445/2010 and A/Ukraine/220/2010 belonged to genetic group 2. These viruses were isolated in the 2009–2010 season, although they slightly differed from other isolates from the same season. Only two of the analyzed isolates obtained in the 2010–2011 season belonged to genetic group 8, namely, A/Ukraine/60/2011 and A/Ukraine/3/2011, and had an A186T specific substitution. Three isolates belonged to genetic group 7. It included two Ukrainian isolates obtained in the 2010–2011 season: A/Ukraine/182/2011, A/Ukraine/130/2011, and one isolate from the 2012–2013 season: A/Zaporizza/417/2013a, as well as the reference strain A/St. Petersburg/100/2011.

All other viruses starting in the 2012–2013 season, to the 2016–2017 season belonged to genetic group 6 and were divided into separate subgroups. Genetic subgroup 6A has not been widely distributed in the world [14], but includes the 2010–2011 isolate A/Ukraine/104/2011 and the 2012–2013 isolates A/Ukraine/3/2013 and A/Ukraine/523/2013. These isolated viruses were characterized by the specific substitutions H138R and V249L. The majority of the 2012–2013 isolates fell within genetic group 6C which became more widespread worldwide during that same season. Our isolates possessed the typical V234I substitution common to this group.

Starting in the 2013–2014 season, the genetic group 6B became widespread not only in Ukraine, but also throughout the world [14]. All viruses from genetic group 6B were characterized by acquired amino acid substitutions D97N, S185T, E47K and S124N. Since the 2015–2016 season, the vast majority of the viruses worldwide has become part of the 6B.1 subgroup according to Nextstrain data [18]. Group 6B.2 is somewhat less numerous, and acquired substitutes are unique to this group, including V152T and V173I in HA1, and E164G and D174E in HA2.

As showcased in Figure 2, in the early 2015–2016 season a division of genetic group 6B into two subgroups occurred. Approximately 63% of viruses belonged to genetic group 6B.1 and around 16%–to genetic group 6B.2. The composition changed later in the 2016–2017 season; the portion of 6B.2 viruses detected dropped to approximately 1% and genetic group 6B.1 viruses became widespread, accounting for 99% worldwide.

### 2.2. Genetic Variability of Pandemic Influenza Viruses

We conducted an analysis of all the substitutions that took place in pandemic influenza viruses isolates in Ukraine from 2009 to 2017 and established their antigenic sites. All viruses were analyzed in comparison with A/California/07/2009 vaccine strain. The H1 molecule HA has five antigenic sites: Sa, Sb, Ca1, Ca2, and Cb; many isolates had mutations in this area that could impact pathogenicity. In this study, we analyzed all genetic changes that occurred in the five antigenic sites of pandemic influenza viruses during seven epidemic seasons. Interestingly, some of the site mutations were conserved and then identified through multiple years. Genetic changes were observed in each of the antigenic sites, but their greatest number was detected on the sites Ca1 and Ca2. Table 2 shows that the smallest number of amino acid substitutions was detected in the antigenic site Cb. In the antigenic site Sa, three substitutions were found. The first amino acid substitution in position 162, where serine substituted for threonine mutation, was initially detected in the 2014–2015 season, only among isolates from the Dnipro region. During the 2015–2016 season, when the 6B viruses were divided into the 6B.1 and 6B.2 genetic groups, all 6B.1 viruses acquired this substitution, which appears in the isolates now. The acquisition of a potential glycosylation site, which may increase the virulence of influenza viruses, determines significance of the S162T substitution [20]. Another replacement was detected in position 163 and had a polymorphism. In 2013–2014, the genetic clade 6B emerged with a featured amino acid substitution from lysine (K) to glutamine (Q) at position 163 (K163Q, H1 numbering) in the hemagglutinin (HA), located at the Sa antigenic site. Only isolate A/Zaporizza/417/2013 was somewhat different as lysine was replaced with isoleucine in position 163 instead of glutamine, as with all other viruses.

In antigenic site Sb, four amino acid substitutions were noticed in three positions. The first substitution S185T was detected among isolates from the 2010–2011 season and appeared consolidated in the virus population at the antigenic level. The adjacent A186T substitution was discovered in two isolates: A/Dnipropetrovsk/445/2010 and A/Ukraine/60/2011, present in different epidemic seasons and in different genetic groups. The third and fourth substitutions include S190G (isolate A/Sumy/797/2009) and S190R (A/Khmelnitsky/88/2016).

The third antigenic site Ca1 had the highest number of substitutions and included five substitutions in four positions. The first substitution, S203T, emerged in the 2009–2010 season, almost immediately after the appearance of pandemic viruses. It was observed in all isolates obtained after the isolate A/Lviv/N6/2009. The next substitution R205K was detected in various epidemic seasons: 2009–2010 and 2014–2015 in isolates A/Ukraine/123/2010 and A/Ukraine/434/2015, belonging to different genetic groups. Also, this substitution was found in reference strain A/Astrakhan/1/2011. The substitution in position 325 was detected in two isolates from different seasons, however, the substituted amino acids were different. The 2010–2011 season isolate A/Ukraine/3/2011 had substitution E235V, whilst the 2014–2015 season isolate had substitution E235D. The last substitution in the antigenic site Ca1 S236P (serine was replaced by proline) was detected in only one isolate A/Zaporizza/631/2016 of the season 2015–2016, belonging to group 6B.1.

The fourth antigenic site of Ca2 also had five amino acid substitutions in four positions. The first amino acid substitution was detected in position 137, where proline was substituted for histidine. This substitution was observed only in one isolate of the 2014–2015 season, belonging to the genetic group 6B. The substitution H138R was also found in the adjacent position 138. Histidine was substituted with arginine in that position. This substitution has been found in two isolates from Ukraine, A/Ukraine/ 523/2012, A/Ukraine/3/2013, as well as in reference strain A/Hong Kong/5659/2012. All of them belonged to the 6B genetic group. The next substitution in the antigenic site Ca2 - A141T was detected only in three isolates from Khmelnitsky (A/Khmelnitsky/675/2016, A/Khmelnitsky/727/2016, A/Khmelnitsky/760/2016) in the 2016–2016 season. The last substitution was found in two variations. The first, D222G, detected mostly in the 2009–2010 season, was found in eight isolates of this season and in one isolate A/Dnipro/580/2016 belonging to the 6B.2 genetic group. The A/Lviv/N6/2009 and A/Ternopil/N11/2009 with D222G substitution resulted in fatalities. Importantly, D222 is located in the receptor-binding pocket at the antigenic site Ca2 HA and may alter the properties of the receptor-binding site. Previously, Shinya et al. showed that substitution D222G in Ca HA causes the transition to a double a-2,3/a-2,6-mediated affinity for the upper and lower respiratory tract epithelium [21]. This mutation was associated with higher replication rates and more severe disease [21]. It has been shown that viruses with such substitutions are found only in fatal and severe cases and are not found in milder forms of infection [22]. Substitution D222N also causes changes in the specificity of the receptors resulting in viral infection of the lower respiratory tract. Supposedly, this substitution is to a lesser extent pathogenic compared with D222G [23]. This substitution was detected in isolate A/Chernivtsi/844/2009, and in the reference strain A/Christchurch/16/2010.

In the fifth last antigenic site Cb, only three substitutions were detected in two positions. The first substitution, A73S, was detected in one isolate A/Kharkiv/963/2016 belonging to the genetic group 6B.1. Two substitutions were found at position 74, S74R (A/Ukraine/130/2011 and A/Odessa/166/2017) and S74N (A/Khmelnitsky/671/2016). The A/Odessa/166/2017 virus with this substitution resulted in fatality and was phylogenetically similar to the viruses from the Maldives (S74R, I295V). In addition, isolate from Odessa possessed an additional unique mutation, T232A. Information about the possible influence of these mutations on virulence of the virus and cause of severe cases is not available.

Figure 3 summarizes the changes in the antigenic sites of influenza viruses, shown as a 3D model of HA. The figure shows substitutions that occurred in antigenic sites of Ukrainian isolates in different colors. Substitutions were observed in each of the five sites. Thus, substitutions in the antigenic sites of the virus were occurring in each epidemic season, some of which could often reoccur in subsequent seasons.

## 3. Discussion

Viruses belonged to 8 genetic groups that circulated worldwide during 2009–2017. Ukrainian isolates within the indicated period belonged to genetic groups 2, 6, 7, and 8. All viruses acquired a number of mutations, which were located in antigenic sites of A(H1N1)pdm09 influenza viruses.

The emergence of S162T and D127E substitutions has led to the acquisition of a potential site for glycosylation, which may increase the virulence of influenza viruses. This has been maintained in the viral population and, in turn, elicits the assumption that the mutation is conducive for infection [10].

By gaining glycosylation sites, virus may escape human immune response. Glycosylation of antigenic sites has been proved to change surface topology of HA glycoprotein. Whereas, virus antigenic properties also change resulting in antibodies failing to recognize the virus [10]. Information about the changes in antigenic sites is very important for prediction of the next dominant strains. The acquisition of carbohydrate side chains by the HA protein is known for the ability to sometimes affect the antigenic properties of the influenza viruses. The carbohydrate chains in or near the antigenic sites may hide them from recognition by antibodies. The substitutions, leading to an acquisition of carbohydrate chains, can create new antigenic variants of influenza viruses [11].

In 2013–2014, the genetic clade 6B emerged with the K163Q (H1 numbering) in HA, in antigenic site Sa. Since then, the 6B clade viruses have diverged further into 6B.1 and 6B.2 subclades, still bearing K163Q. The 6B.1 subclade has predominated globally since the 2015–2016 season [24]. Viruses from 6B, 6B.1 and 6B.2 genetic groups were antigenically similar to A/California/07/2009 vaccine strain, as showcased by post-infection antisera [3,6]. Vaccine effectiveness for 6B.1 subgroup viruses in adults born in 1958–1979 was significantly lower compared to that in other age groups (22 and 61% accordingly) [25]. A hypothesis has been suggested that middle-age people previously infected by seasonal influenza A(H1N1) viruses may be immune to new pandemic A(H1N1)pdm09 viruses that have acquired K163Q substitution [26].

The S185T substitution in antigenic site Sb falls within a domain defining the receptor-binding site (RBS). It is well documented that amino acid changes near or in the RBS may have influence on the antigenic properties of pandemic influenza viruses A(H1N1)pdm09. The substitution in antigenic site Sb A186T was also in the RBS [27].

Importantly, substitution D222G is located in the receptor-binding pocket in the antigenic site Ca HA and may alter the properties of the RBS [21]. In addition, the K163I substitution observed in A/Zaporizza/417/2013 may impart altered antibody-binding properties [28,29].

Ukrainian isolates retained similarity to the vaccine strain A/California/07/09 during 2009–2017, despite the observed mutations present in the antigenic sites. However, after the emergence of the new genetic groups 6B.1 and 6B.2, WHO recommended a new vaccine strain with changed antigenic properties A/Michigan/45/2015 [30].

In this paper, we identified amino acid substitutions that occurred in isolates of pandemic influenza viruses isolated in Ukraine from 2009 to 2017, and analyzed their possible impact on virus virulence. The genetic characterization and surveillance of circulating influenza strains are extremely important for the evaluation of vaccine effectiveness, and compositing annual influenza vaccine development.

## 4. Materials and Methods

### 4.1. Sample Collection

This study used clinical samples of nasopharyngeal, pharyngeal, nasopharynx, nose swabs, and autopsy material collected in the first three days and no later than the fifth day of illness from patients with suspected influenza and SARI (severe acute respiratory infections). Autopsy material from dead bodies was obtained from the Regional Department of Morbid Anatomy and Office Forensics.

Samples were collected during routine diagnosis of influenza-like illness (ILI) in patients from sentinel clinics in four cities in Ukraine (Kyiv, Dnipro, Odesa, and Khmelnytsky). All the clinical samples were collected under bioethical guidelines research among human beings for human subjects research under Institutional Review Board approval #281 (11/01/2002); approving authority: Ministry of Health of Ukraine.

### 4.2. Extraction of Nucleic Acids and RT-PCR

Virological and molecular genetic methods were used to confirm influenza cases and analyze genetic sequences of isolates. Laboratory diagnosis was conducted by real-time PCR (polymerase chain reaction). To detect influenza virus types A and B, the following extraction kits were used: InnuPREP Virus DNA/RNA, Analytik Jena, Reinach, Switzerland, and primers and probes for respective markers: Univ inf A, sw A, sw H1, H1, H3, Univ inf B, RNase P (Biosearch Technologies, Petaluma, CA, USA). Molecular methods were carried out in accordance with the protocol provided by the WHO Collaborating Centre for Influenza at Atlanta’s Centers for Disease Control and Prevention (CDC) (version 30 April 2009) [31]. The Thermo Scientific Verso 1-Step qRT-PCR ROX Kit (Waltham, Massachusetts, USA) was used for RT-PCR [31].

### 4.3. Virus Isolation, Sequencing and Genetic Analysis

Viral isolation and cultivation included passages in the MDCK cell line (epithelial kidneys cells from a Cocker Spaniel) obtained from CDC [32]. Isolates were later used for strain identification and sequencing. The sequencing of influenza virus isolates was performed by the World Influenza Center in London using RNA-SEQ technology. The influenza virus isolates in a cell culture medium were frozen in liquid nitrogen (–80 °C) and shipped following the cold chain principle. The sequences of influenza viruses from other countries were obtained from the GISAID (Global Initiative on Sharing All Influenza Data; http://platform.gisaid.org) website using BLAST analysis. Sequences were aligned using the ClustalW algorithm. The neighbor joining method applying Kimura 2-parameter model built phylogenetic trees. Nucleotide sequences were translated into amino acid sequences using MEGA 7 software [33]. The 3D structures were constructed in Chimera 1.11.2rc [34] software using crystallographic structures of HA (PDB files) from the Protein Data Bank [35]. Spread of A(H1N1)pdm09 influenza virus of genetic groups 6B.1 and 6B.2 was taken from Nextstrain https://nextstrain.org [18,19].

## Figures and Tables

**Figure 1 pathogens-08-00194-f001:**
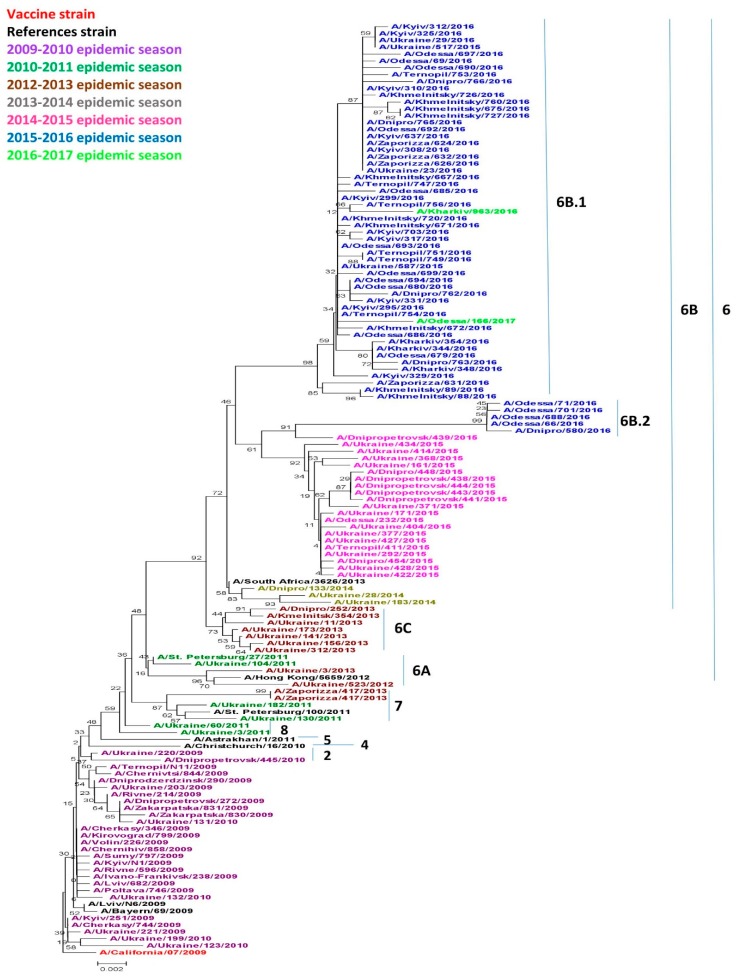
Phylogenic comparison of A(H1N1)pdm09 influenza virus in 2009–2017 based on oligonucleotide sequences of hemagglutinin (HA) conducted by the Neighbor Joining method, Kimura 2-parameter model with 1,000 bootstrap replications.

**Figure 2 pathogens-08-00194-f002:**
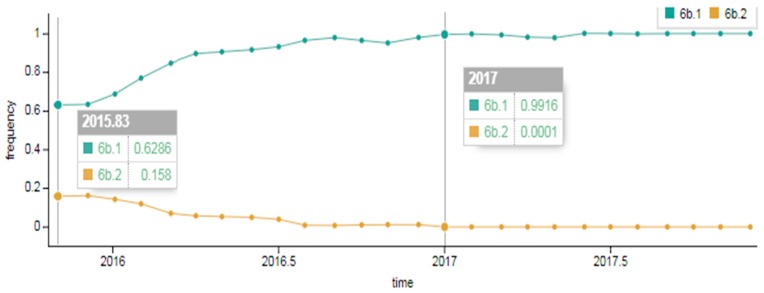
Spread of A(H1N1)pdm09 influenza virus of genetic groups 6B.1 and 6B.2 in the early 2015–2016 season and in the late 2016–2017 season in the world (data provided by NextFlu – now Nextstrain https://nextstrain.org/) [18,19].

**Figure 3 pathogens-08-00194-f003:**
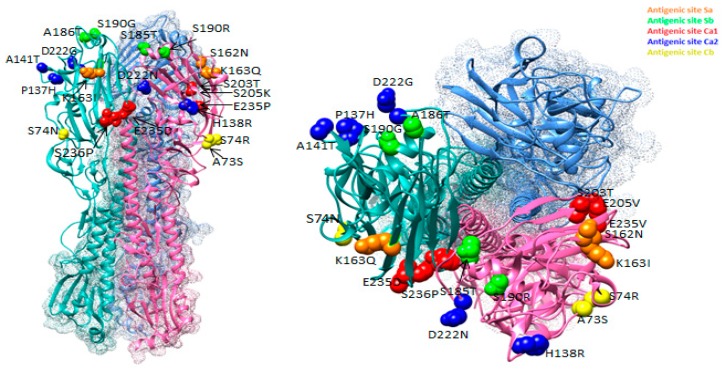
Location of amino acid substitutions in antigenic sites of HA viruses of A(H1N1)pdm09 influenza discovered among isolates isolated in Ukraine between 2009 and 2017 (PDB ID–3LZG).

**Table 1 pathogens-08-00194-t001:** Mutations in genetic groups 1–8 of A(H1N1)pdm09 with the specified reference strain belonging to the genetic group. The mutations in HA1 and HA2 are outlined, with mutations in HA2 highlighted in blue font.

Group	Mutations	Reference Strain
1		A/California/07/2009 *
2	N31D, S162N (+CHO), A186T in HA1, E47K and S124N in HA2	-
3	A134T and S183P in HA1	A/Hong Kong/3934/2011
4	N125D in HA1 and E47K in HA2	A/Christchurch/16/2010
5	D97N, R205K, I216V, V249L in HA1 and E47K in HA2	A/Astrakhan/1/2011
6	D97N, S185T in HA1, E47K and S124N in HA2	A/St. Petersburg/27/2011
6A	H138R, V249L in HA1	A/St. Petersburg/27/2011
6B	D97N, S185T, S203T K163Q, A256T and K283E in HA1, and E47K, S124N, E172K in HA2	A/Michigan/45/2015*
6B.1	S84N, S162N (+CHO) and I216T in HA1	A/Michigan/45/2015*
6B.2	V152T and V173I in HA1 and E164G and D174E in HA2	-
6C	V234I in HA1; some viruses also have substitutions V30A and A186T in HA1;	-
7	S143G, S185T, A197T in HA1, E47K and S124N in HA2	A/St. Petersburg/100/2011
8	A186T, V272A in HA1, E47K, N146D, T147K and V193A in HA2	-

Note: * Vaccine strains.

**Table 2 pathogens-08-00194-t002:** Amino acid substitutions in antigenic sites of A(H1N1)pdm09 influenza viruses within the period between 2009 and 2017.

Antigenic Site	Substitution	The Name of the Isolate in Which the Substitution Was Discovered	Number of Isolates in Each Group	Total Number of Isolates Identified
**Sa**	S162T	A/Dnipropetrovsk438,441,443,444,448/2015	5	136
	S162T	All isolates from group 6B.1	55	
	K163Q	All isolates beginning with A/Ukraine/312/2013	91	
	K163I	A/Zaporizza/417/2013	1	
**Sb**	S185T	All isolates beginning with A/Ukraine/3/2011	93	
	A186T	A/Dnipropetrovsk/445/2010, A/Ukraine/60/2011	2	
	S190G	A/Sumy/797/2009	1	
	S190R	A/Khmelnitsky/88/2016	1	
**Ca1**	S203T	All isolates beginning with A/Lviv/N6/2009	123	
	R205K	A/Ukraine/123/2010, A/Astrakhan/1/2011, A/Ukraine/434/2015	3	
	E235V	A/Ukraine/3/2011	1	
	E235D	A/Ukraine/368/2015	1	
	S236P	A/Zaporizza/631/2016	1	
**Ca2**	P137H	A/Ukraine/414/2015	1	
	H138R	A/Ukraine/523/2012, A/Ukraine/3/2013, A/Hong Kong/5659/2012	3	
	A141T	A/Khmelnitsky/675,727,760/2016	1	
	D222G	A/Kyiv/251/2009, A/Cherkasy/744/2009, A/Lviv/N6/2009, A/Sumy/797/2009, A/Rivne/596/2009, A/Volin/226/2009, A/Zakarpatska/831/2009, A/Ternopil/N11/2009, A/Dnipro/580/2016	9	
	D222N	A/Chernivtsi/844/2009, A/Christchurch/16/2010	2	
**Cb**	A73S	A/Kharkiv/963/2016	1	
	S74R	A/Ukraine/130/2011, A/Odessa/166/2017	2	
	S74N	A/Khmelnitsky/671/2016	1	

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
