# Peer review of "Antigenic Site Variation in the Hemagglutinin of Pandemic Influenza A(H1N1)pdm09 Viruses between 2009–2017 in Ukraine"

_pathogens, 2019, doi:10.3390/pathogens8040194_

Round 1
Reviewer 1 Report
The manuscript that I reviewed “Antigenic Site Variation in the Hemagglutinin of Pandemic Influenza A(H1N1)pdm09 Viruses Between 3 2009-2017 in Ukraine” is a study aimed to perform a genetic and phylogenetic analysis of hemagglutinin (HA) influenza isolates from Ukraine from 2009 to 2017, including the vaccine strain and reference strains to identify the genetic changes that occurred in the five antigenic sites Sa, Sb, Ca1, Ca2, and Cb, subjected to antibody-mediated immune pressure, and that represent the basis of the evolutionary dynamics observed in the influenza viruses. Among eight, etween 2009 and 2017 the Ukrainian isolates clustered in the influenza genetic groups 2, 6, 7, and 8. Genetic changes were observed in each of the antigenic sites. Despite the study demonstrate a genetic variability of HA protein, the A(H1N1)pdm09 strains isolated in 2009-2017 in Ukraine showed similarity to the vaccine strain A/California/07/09 during 2009-2017.
General comments
Overall the study conducted is interesting, detailed and supported by presented data. The manuscript is well written as a whole, and I will not criticize at all, out of some little observations.
1)I suggest to the Authors that it will be useful to enrich the introduction section of some just more notions about the etiology of the influenza viruses with particular regard to the genes and the encoded proteins. Furthermore, they could add a sentence to better describe the HA molecule to introduce and explain the HA1 and the HA2 subunits.
2)Line 60: I suggest to the Authors to replace “that resulted” with “resulting”.
3)Line 61: I suggest to rewrite the sentence “Viruses belonged to 8 genetic groups that circulated worldwide at the time.” with “The viruses circulating worldwide at that time belonged to 8 genetic groups”.
Author Response
Dear reviewer,
Thank you very much for your revision and comments.
We have updated the Introduction section, added general information about influenza genes and impact of amino acid substitution on the virus antigenic properties. In addition, we have described the structure of HA and HA1, HA2 subunits. According to your comment, we have replaced “that resulted” with “resulting” in the Line 60. We have rewritten the sentence (Line 61): “Viruses belonged to 8 genetic groups that circulated worldwide at the time” with “The viruses circulating worldwide at that time belonged to 8 genetic groups”.
Thank you very much for your help!
Best regards,
Oksana Zolotarova.
Reviewer 2 Report
Due to the propensity of influenza viruses to undergo antigenic drift, active influenza surveillance and genetic characterization of circulating strains is crucial to understanding vaccine effectiveness and to inform annual influenza vaccine compositions. The manuscript by Zolotarova et. al, nicely characterizes hemagglutinin antigenic site variation of pandemic influenza A (H1N1)pdm09 virus isolates collected from 2009-2017 in Ukraine. The text could be significantly improved by proofreading and proper sentence structure, as there are grammatical errors throughout. It is suggested that the authors have a colleague or a consultant versed in the use of English read and edit the manuscript.
Specific comments:
(1) Suggest expanding the introduction to include a few sentences that describe HA structure and define HA1 and HA2 domains, since the genetic analysis specifies mutations in both domains.
(2) Line 38: this line is confusing as written. Did you mean to say "...to include only actual 'circulating' strains?
(3) Lines 134-141: The writing is awkward here. This should be re-organized into one paragraph.
(4) Lines 188-190: What is antigenic site Ca HA? Did you mean Ca1 or Ca2?
(5) Line 215: Suggest adding a lead-in sentence to recap the study objective. For example, "Phylogenetic and genetic analyses were performed on Ukrainian pandemic influenza virus isolates collected between 2009-2017 to identify trends in mutation locations and amino acid substitutions."
Author Response
Dear reviewer,
Thank you very much for your work and comments.
We asked English-speaking professional to review English language and writing style, and implemented the edits into the manuscript.
According to your comments:
We have updated the Introduction section with additional information about HA structure and HA1, HA2 domains. We have changed line 38 and added phrase: actual “circulating” strains. We have reorganized lines 134-141 into one paragraph. In the lines 188-190 Ca2 domain was implied, and added accordingly. We have added recommended sentence to the Introduction section.
Thank you very much for your help!
Reviewer 3 Report
This manuscript presented good data however, it was difficult to follow since the results and discussion read like a catalog of mutations. The authors could enhance their paper and increase it's significance if they could link these mutated isolates and diverged populations with clinical features, pathogenicity, casualties etc. It would ne nice if the authors could indicate which strains (if any) cause more pathology than others (human or animal). The authors briefly discus mutations in antigenic sites possibly impacting vaccine efficacy. If they were to expand this and tie to their data they could increase the significance of their findings. The manuscript ended abruptly. The authors need to include a few sentences in which they conclude the manuscript.
Author Response
Dear reviewer,
Thank you very much for your revision and comments.
We asked English-speaking professional to review English language and writing style, and implemented the edits into the manuscript.
Unfortunately, very limited information on clinical features, pathogenicity and casualties is available. In addition, we do not have experimental data about influence of antigenic sites changes on vaccine effectiveness in Ukraine; our statement is based on literary sources data.
We have information about fatal cases. In the Results section, we have mentioned about A/Odessa/166/2017 strain with changes in antigenic site Cb, which caused death; A/Lviv/N6/2009 and A/Ternopil/N11/2009 with D222G substitution resulted in fatalities.
In the Discussion section, we included several additional sentences to conclude the manuscript properly.
Thank you very much for your help!